# Non-Steroidal Anti-Inflammatory Drugs in Colorectal Cancer Chemoprevention

**DOI:** 10.3390/cancers13040594

**Published:** 2021-02-03

**Authors:** Jadwiga Maniewska, Dagmara Jeżewska

**Affiliations:** Department of Medicinal Chemistry, Faculty of Pharmacy, Wroclaw Medical University, Borowska 211, 50-556 Wrocław, Poland; dagmara.jezewska@student.umed.wroc.pl

**Keywords:** colorectal cancer, cancer chemoprevention, NSAIDs, clinical trials, COX-2

## Abstract

**Simple Summary:**

There is growing evidence from epidemiologic, preclinical and clinical studies suggesting that non-steroidal anti-inflammatory drugs (NSAIDs) play a beneficial role in colorectal cancer chemoprevention. They reduce the risk of colorectal polyps, mostly by cyclooxygenase-2 inhibition. The aim of our work was to describe the current state of scientific knowledge on the potential added value of the use of NSAIDs (such as aspirin, sulindac, and celecoxib) as chemopreventive agents in patients at risk of colorectal cancer. The study confirmed that there is a link between the long-term use of the NSAIDs and a decrease in the risk of colorectal cancer.

**Abstract:**

Since colorectal cancer is one of the world’s most common cancers, studies on its prevention and early diagnosis are an emerging area of clinical oncology these days. For this study, a review of randomized controlled, double-blind clinical trials of selected NSAIDs (aspirin, sulindac and celecoxib) in chemoprevention of colorectal cancer was conducted. The main molecular anticancer activity of NSAIDs is thought to be a suppression of prostaglandin E_2_ synthesis via cyclooxygenase-2 inhibition, which causes a decrease in tumor cell proliferation, angiogenesis, and increases apoptosis. The lower incidence of colorectal cancer in the NSAID patients suggests the long-lasting chemopreventive effect of drugs studied. This new approach to therapy of colorectal cancer may transform the disease from a terminal to a chronic one that can be taken under control.

## 1. Introduction

Colorectal cancer (CRC) is one of the most common human malignancies in the Western countries, being the world’s fourth most deadly malignancy [1]. An unhealthy diet and lifestyle, common in well-developed countries, are implicated as risk factors for CRC [2,3,4,5,6,7]. Chronic inflammation is also recognized as a potential risk factor for tumor development. Therefore, targeting inflammatory pathways has proven effective in preventing the formation of colon tumors and their malignant progression in both preclinical and clinical studies [8,9,10,11,12]. Moreover, the inflammation and the angiogenesis are host-dependent cancer hallmarks that can be targeted using preventive approaches long before tumors initiate and progress [13,14].

Unfortunately, the screening programs that demonstrated very high value in many types of cancer entail early diagnosis but do not prevent tumor development. Moreover, colonoscopy represents an effective CRC screening option but is an expensive procedure with a relatively low patient acceptance rate (although it may reduce mortality). That is why the modification of lifestyle and dietary factors to reduce the incidence of cancer has been lately strongly promoted and chemoprevention studies have grown in importance [15,16,17,18,19].

The term chemoprevention was introduced 40 years ago by Michael B. Sporn. It was defined as the use of natural, synthetic or biological agents to reverse, suppress or prevent either the initial phases of carcinogenesis or progression of premalignant cells to invasive disease [20,21,22,23]. Remarkably important is that chemoprevention is not aimed at replacing the role of surgery and chemotherapy itself. It is instead a kind of adjuvant therapy, involving disruption of a variety of steps in tumor initiation, promotion, and progression [24,25]. A few published randomized trials have shown that chemopreventive therapies may be effective against colorectal cancer [26,27,28,29,30]. Nevertheless, there is still the need to develop guidelines for the management of patients with a higher risk of colorectal cancer or the early treatment with adjuvant chemoprevention therapy.

## 2. Review

### 2.1. Cyclooxygenase in Tumor Genesis

Prostaglandin endoperoxide synthase (PGHS), commonly known as cyclooxygenase (COX), is a dimeric membrane enzyme converting arachidonic acid (derived from membrane phospholipids by phospholipase A_2_) to prostaglandin H_2_. Prostaglandin H_2_ is a precursor of other prostaglandins and thromboxanes, playing a major role in mediating inflammation, gastric cytoprotection, and platelet aggregation. The synthesis of prostaglandin H_2_ is a two-step process, occurring at spatially separated active sites of the enzyme: a cyclooxygenase and a peroxidase site. In the first step, arachidonic acid is oxygenated at the COX active site to form hydroperoxide prostaglandin G_2_, which is then reduced at the peroxidase active site to the alcohol—prostaglandin H_2_ (Figure 1). NSAIDs, which bind to the COX active site instead of arachidonic acid, are competitive binding inhibitors of the enzyme. There are three known isoforms of the prostaglandin H_2_ synthase, out of which the main ones are COX-1—the constitutive isoform and COX-2—the inducible isoform of the enzyme [31,32]. In the last two decades, many research results showing the link between the overexpression of COX-2 and the occurrence of many human malignancies, for example, colorectal, breast, pancreatic, and lung cancer, have been published. The data revealed that COX-2 plays a role in different steps of cancer progression and metastasis formation [26,33,34].

### 2.2. NSAIDs

NSAIDs are mostly weak organic acids with hydrophobic properties that determine their binding ability to the membrane protein COX. NSAIDs are well known for their antipyretic, analgesic, and platelet antiaggregant effects [38]. The most popular examples of the NSAID family are listed in Table 1. NSAIDs inhibit COX enzyme activity which inhibits prostanoid biosynthesis (Figure 1).

As bioactive lipids, prostanoids activate specific cell membrane receptors (e.g., prostaglandins, prostacyclin I_2_, and thromboxane A_2_ receptors) related to specific biological functions, therefore, inhibition of their synthesis is associated with the occurrence of adverse clinical effects after the long-term use of NSAIDs [39]. COX inhibition can increase the risk of gastrointestinal ulcers and bleedings, it also interfere with kidney function [40]. The selective COX-2 inhibitors (such as celecoxib and rofecoxib) were intended to lower the rates of gastrointestinal adverse events but are associated with cardiovascular risk (e.g., stroke or heart failure) [41]. One of the proposed mechanisms for this risk of NSAIDs is the observed shift in the prothrombotic-antitrombotic balance on endothelial surface towards thrombosis. The simplified hypothesis of an association between the degree of COX-2 inhibition and the cardiovascular risk is that the more COX-2 inhibition that an NSAID exerts relative to COX-1 inhibition, the higher the risk of cardiovascular events is [41]. Because of increased risk of vascular events, long-term use (more than 5 years) of COX-2 inhibitors in prevention is rather not feasible (more feasible is aspirin use) [42]. Low-dose aspirin (COX-1 inhibitor) therapy inhibits TXA_2_-dependent platelet function, that is why patients are taking it to prevent myocardial infarction, but antiplatelet effect of this drug might also increase the risk of bleeding during a subsequent operation [43]. NSAIDs may also impair the therapeutic benefits of ACE inhibitors and β-blockers by inhibiting prostaglandins biosynthesis in the kidneys, which results in impaired vasodilatation, decreased renal function, sodium and water retention which may cause edema [39].

### 2.3. Cancer-Related Inflammation

The suggestion that inflammation might be linked to cancer was firstly made by the German physician Rudolf Virchow in the late nineteenth century [44]. He noticed that malignant tumors arise at regions of chronic inflammation and contain inflammatory infiltrates. Recent studies have shown that chronic inflammation increases the risk of both tumor development and progression [45,46,47].

Chronic inflammation may predispose to some cancers, while tumors already formed sustain the inflammatory process that stimulates cancer progression. This phenomenon is like a self-propelling mechanism. It is estimated that chronic inflammation is responsible for about 20% of all cancers. The activated transcription factors are present in the inflammatory microenvironment. They stimulate the expression of various cytokines, chemokines, COX-2, and, consequently, prostaglandins (inflammatory mediators). As a result of these pro-inflammatory substances being released, inflammatory cells accumulate in the surrounding tissue. As a result of these pro-inflammatory substances being released, inflammatory cells accumulate in the surrounding tissue. The transcription factors get activated, and the secretion of cytokines proceeds. The inflammatory process intensifies [45]. In the inflammatory environment, there are also reactive forms of oxygen and nitrogen present [48].

Moreover, changes in the expression of microRNA occur, which most likely contribute to the carcinogenesis [49]. In the experimental animal and human models, it has been proven that inflammatory cells characteristic of the inflammatory microenvironment (i.e., chemokines and cytokines) are also present in the microenvironment of all cancers from the earliest stages of their development. NF-κB, STAT3, and the major pro-inflammatory cytokines of the interleukin family, i.e., IL-1β, IL-6, IL-23, and TNF-α (tumor necrosis factor-alpha), are considered the key endogenous cancer-related inflammatory factors. NF-κB is a major endogenous promoter that supports carcinogenesis on many levels. It activates the expression of genes encoding pro-inflammatory cytokines, enhances the expression of cyclooxygenase-2, the inducible nitric oxide synthase, and various other angiogenesis-stimulating factors. Moreover, it activates anti-apoptotic genes, e.g., BCL-2, leading to the so-called cancer cell immortality. NF-κB was also determined to promote metastasis. Therefore, there is strong evidence that the inflammatory environment supports all three stages of cancerogenesis, from its onset until the disease progression [45]. The relationship between inflammation and cancer is best documented for the gastrointestinal tract, where the link between the organ affected by chronic inflammation and subsequent neoplasia is thus clearly visible. Multiple examples include colorectal cancer that can develop as a result of Crohn’s disease or ulcerative colitis, chronic Helicobacter pylori infection increasing the risk of gastric cancer, reflux disease predisposing to esophageal cancer [48], and viral hepatitis (type B and C) that promotes liver cancer [50,51].

### 2.4. The Concept of Cardio-Oncology

Chronic inflammation participates in the pathogenesis of both cancer and cardiovascular disease. It may determine and aggravate immuno-senescence with formation of abnormal medullary clones of immune cells with altered function. The emergence of clonal hematopoiesis as a common risk factor for cardiovascular disease and for hematologic malignancies links an atherosclerosis and cancer with the inflammation as a driver. Inflammatory cell infiltrate burden usually associates with a poor prognosis in patients with cancer. The inflammatory biomarkers are also used in the diagnosis of cardiovascular disease [52]. The premalignant state, for hematologic cancer, termed clonal hematopoiesis of indeterminate potential (CHIP) seems to be associated with increase in cardiovascular risk independent of LDL cholesterol level, hypertension, or diabetes. Due to this, Libby et al. suggest that careful co-ordination between practitioners of cardiovascular medicine, hematology and oncology is needed [53]. Moreover, CHIP occurrence is age dependent—up to 20% of septuagenarians have it. Individuals found to have this condition require expert management of their cardiovascular risk [52]. The possibility of preventive use of NSAIDs (e.g., long-term use of aspirin) in these patients requires further investigation.

### 2.5. NSAIDs in Colorectal Cancer Prevention

NSAIDs have been one of the most promising agents in the chemoprevention of colorectal cancer [54,55,56,57]. They are a widely used medication with the well–known molecular target. Their activity involves the inhibition of COX enzymes [58]. NSAIDs are widely used to relieve pain, reduce inflammation and fever. A low-dose aspirin therapy (75 mg per day) has also proven to reduce the risk of stroke and heart attack effectively [59]. Recently, however, the COX-1 inhibition, which leads to suppression of platelet activation, facilitates immunosurveillance, and prevents the hematogenous spread of malignancy, has been suggested as another putative mechanism of cancer prevention [60].

Nevertheless, the main anticancer activity of NSAIDs is thought to be a suppression of prostaglandin E_2_ synthesis via COX-2 inhibition, which causes a decrease in tumor cell proliferation, angiogenesis, and increases apoptosis [36,61]. Although many of the anticancer mechanisms of NSAIDs are defined as COX-dependent, several signal transduction pathways (e.g., including nuclear factor-kappa B, NF-κB) have been confirmed as COX-independent NSAID-induced effects [62,63,64,65]. The correlation between the COX expression and colorectal cancer, including prognostic factors and putative chemopreventive agents, has been widely studied worldwide and reviewed in the past [66].

Most likely, the anticancer activity of NSAIDs will be due to the disruption of the inflammatory microenvironment. Mechanisms involved in the chemopreventive action of NSAIDs can be divided into COX-dependent and COX-independent. NSAIDs block COX-2 and thus may inhibit its adverse effect on carcinogenesis. The role of COX-2 in the process of carcinogenesis has been described in Section 2.3. Therefore, mechanisms independent of COX-2 are presented here. Wnt/β-catenin may play an essential role in the signaling pathway [67]. Wnt pathway aberrations are observed in various types of cancers. The Wnt signaling pathway leads to the transfer of β-catenin from the cytosol to the nucleus. β-catenin is a protein that controls the activity of the T-cell transcription factor regulating, among others, cell proliferation, differentiation, and survival [68]. NSAIDs, such as sulindac and celecoxib, can reduce concentrations of nuclear β-catenin and trigger its degradation, which would support their proapoptotic and antiproliferative effects. Sulindac sulfone exhibits similar activity, which confirms its COX-independent mechanism [69].

NSAIDs can also independently of COX-2 inhibit the activation of the NF-κB transcription factor that contributes to increased proliferation and promotes cancer cells’ survival. The IκB inhibitor protein binding regulates the NF-κB activity. In a bound form, the NF-κB remains inactive. Aspirin (ASA) is a competitive IκB kinase inhibitor [70] that releases the NF-κB from the inactive protein inhibitor complex. Thus, ASA prevents the initiation of transcription of many pro-inflammatory and pro-carcinogenic factors [71]. NSAIDs may also interact with PPARs (peroxisome proliferator-activated receptors) that are transcription factors. The main task of PPAR is to regulate the metabolism of fatty acids and control the concentration of glucose in the body. In contrast, the isotype γ of PPAR may play a role in cell proliferation, differentiation, and survival. Activation of PPARγ may result in the induction of apoptosis, inhibition of angiogenesis, and the antiproliferative effects [68]. Ligands that activate PPARγ include NSAIDs, such as ibuprofen, sulindac sulfide, and indomethacin [70]. Conversely, NSAIDs suppress transcription of the isotype δ of PPAR, which is found beneficial as prostanoid-activated PPARδ contributes to the stimulation of tumor growth, especially in the case of the large intestine. NSAIDs can also induce apoptosis by increasing the activity of 15-lipooxygenase 1 (15-LOX-1) [33].

### 2.6. NSAIDs’ Effect on the Gut Microbiome

Bacteria in the human gastrointestinal tract have an important role in the formation of colorectal cancer [72]. Gut bacteria also reflect the types of medication that people ingest. It was found that NSAIDs users exhibited a different gut microbiome profile than nonusers. The types (not the number) of medications used caused the greatest difference in microbiome [73]. It was also shown (in humans) that aspirin may inhibit the growth of pro-inflammatory bacteria in a dose-dependent manner. Prizment et al. [72] suggested that aspirin alters the composition of gut microbiome in a way consistent with decreased inflammation and reduced colorectal cancer. They found that aspirin induces changes in the gut microbiome. The drug changed several bacterial taxa in a way consistent with reduced colorectal cancer [73].

### 2.7. Clinical Trials

Clinical trials have examined several drugs and micronutrients for their potential effects on colorectal cancer-associated inflammation. Analysis of epidemiological studies revealed that NSAIDs such as aspirin, sulindac, and celecoxib (selective COX-2 inhibitor) might reduce the risk of CRC.

For this study, a review of the efficacy of NSAIDs in the chemoprevention of colorectal cancer was performed by searching the MEDLINE/PubMed (https://pubmed.ncbi.nlm.nih.gov) and Cochrane Central Register of Controlled Trials (CENTRAL) databases (https://www.cochranelibrary.com/central). The review was limited to randomized controlled, double-blind clinical trials (Table 2), as the evidence from such trials is objective and considered most reliable. Moreover, only the clinical trials involving NSAIDs in monotherapy were considered to determine the significance of the drug alone in the overall chemopreventive effect.

#### 2.7.1. Acetylsalicylic Acid in Chemoprevention of Colorectal Cancer

Clinical studies on aspirin in cancer chemoprevention focused on evaluating its efficacy in preventing colorectal cancer. Up to date, seven extensive experiments on chemoprevention with ASA in monotherapy and polytherapy with folic acid or eicosapentaenoic acid, and ASA with starch were conducted. We focused only on the use of ASA alone or ASA with starch. One was the Colorectal Adenoma Prevention Study (CAPS) in patients with a history of colorectal cancer. Eligible patients could not have had a relapse within at least five years after tumor resection. Individuals suffering from hereditary colorectal cancer syndromes (Familial Adenomatous Polyposis–FAP and Hereditary Nonpolyposis Colorectal Cancer or HNPCC) were not eligible for the study. It recruited a total of 635 patients aged 30–80 years who had a colonoscopy with polypectomy within four months before the intervention started. Subjects were randomized; 317 patients were assigned to the group treated with aspirin at a dose of 325 mg/day and 318 patients—to the group receiving placebo. The clinical endpoint was adenoma incidence in the large intestine. The first control colonoscopy was performed about one year from the beginning of the intervention, and the results were obtained for 517 patients. At least one adenoma was detected in 43 out of 259 (17%) and in 70 out of 258 (27%) patients from the study and the control group, respectively. Also, the average number of adenomas detected was lower in the group receiving aspirin. The corrected relative risk of adenoma recurrence in the study group compared to the control group was 0.65 (95% CI, 0.46–0.91). The type and the number of adverse events occurring during the trial were similar in both groups. The results of the experiment indicate that the long-term use of aspirin at a dose of 325 mg per day may significantly reduce the risk of adenoma recurrence in individuals with a previous medical history of colorectal cancer [74].

In the period 1996–2005, the Association pour la Prevention par l’Aspirine du Cancer Colorectal (APACC) study was conducted, evaluating the effect of ASA administered as water-soluble sachets with lysine acetylsalicylate (acetylsalicylic acid salts with better solubility) on the development of colorectal adenomas. A total of 272 patients were enrolled, 73 of whom received the study drug at a dose of 160 mg/day, 67 patients received 300 mg/day, and 132 participants received a placebo. Inclusion criteria included at least one adenoma >0.5 cm in diameter or at least three adenomas of any size. All lesions had to be removed within three months before the start of the intervention. Patients with hereditary syndromes, i.e., FAP and HNPCC, after bowel resection and past colorectal cancer, were excluded from the recruitment process. The APACC study assumed treatment continuation for four years. The endpoint was defined as the percentage of patients with a relapse confirmed by colonoscopy after one year and after four years from the beginning of the study. Recurrence was defined as the detection of at least one adenoma in the large intestine. One year after starting the study, the colonoscopy results were obtained from 238 patients. At least one adenoma was observed in 38 out of 126 (30%) patients receiving aspirin, and in 46 out of 112 (41%) patients from the placebo group RR = 0.73 (95% CI, 0.52–1.04). In the fourth year, a colonoscopy was performed for 185 participants. The percentage of participants with a relapse did not significantly differ in both groups. At least one adenoma was confirmed by colonoscopy in 42 out of 102 (41%) patients on aspirin and 33 out of 83 (40%) patients in the placebo group. Also, no significant differences were noted in the incidence of advanced adenomas between the groups. Advanced lesions occurred in 10% and 7% of patients from the treatment and control groups, respectively. Although favorable results were observed one year after starting the aspirin treatment, finally, in the fourth year, its chemopreventive effect on the prevention of colorectal adenomas recurrence was not confirmed [75,82].

Another study was conducted on an Asian population, where 311 patients aged 40–70 years were qualified. The inclusion criteria included one or more adenomas and/or adenocarcinomas restricted to the colorectal mucosa. The lesions had to be removed before the start of the two-year intervention. The treatment group consisted of 152 patients that received ASA as enteral tablets at a dose of 100 mg per day, while the control group included 159 participants on placebo. The primary endpoint was the frequency of adenoma or adenocarcinoma recurrence in both groups, expressed as the odds ratio (OR). In the treatment group, 96 out of 152 (63%) patients did not present any relapse, while in the control group, 86 out of 159 (54%) patients showed no evidence of relapse. The corrected OR was 0.60 (95% CI 0.36–0.98). It is worth noting that the risk of relapse was significantly lower among participants who did not smoke. For non-smoking patients receiving ASA, the odds ratios were 0.37 (95% CI, 0.21–0.68) [76].

The Colorectal/Adenoma/Carcinoma Prevention Programme 2 (CAPP2) is a 2 × 2 factorial design study involving patients with Lynch syndrome (HNPCC), the most common cause of hereditary colorectal cancer. The experiment studied the effect of aspirin 600 mg/daily (2 × 300 mg) or resistant starch 30 g/daily for up to four years. However, only data and results related to the application of ASA are presented here. The study is unique compared to the previous ones in terms of the endpoint, being the incidence of colorectal cancer, and the long-term follow-up period. A total of 861 patients were randomized, of whom 427 were assigned to the aspirin-treated group and the remaining 434 patients to the placebo group. The intervention started between the years 1999 and 2005. The average time of intervention with ASA was 25 months, while the average follow-up time exceeded seven years. Up to this point, since the randomization, 98 patients are known to have developed colorectal cancer—40 (9%) patients from the treatment group and 58 (13%) patients from the control group. Results of the intention-to-treat analysis (ITT) indicate a significant reduction in the risk of colorectal cancer in the group receiving aspirin HR = 0.65 (95% CI, 0.43–0.97). When the data set is narrowed to patients who have been using the intervention for more than two years (population according to the clinical trial protocol, PP), the HR value is 0.56 (95% CI, 0.34–0.91). Additionally, the per-protocol analysis showed that treatment with ASA 600 mg reduces the risk of other Lynch syndrome-related cancers (HR 0.63; 95% CI, 0.43–0.92). No significant differences between the groups in the incidence of adverse events have been reported [77].

Furthermore, the effect of long-term aspirin use on the risk of cancer (including colorectal cancer) was assessed in the Women’s Health Study (WHS). It was a large-scale research study that recruited a total of 39,876 women from the United States. The participants were at least 45 years old and were generally in good health. The exclusion criteria included a history of past cancer, cardiovascular diseases, and other serious chronic diseases. Patients were randomly assigned to a treatment group receiving ASA 100 mg every other day (*n* = 19,934) or to a group receiving placebo (*n* = 19,942). The intervention lasted for an average of ten years, with the primary endpoint being the number of all cancer incidence cases in the study group relative to the control group. There were no significant differences between the groups—1438 and 1427 cases of invasive cancer (excluding non-melanoma skin cancer) were confirmed respectively in the treatment and control groups RR = 1.01 (95% CI, 0.94–1.08). There was also no positive effect noted in reducing the risk of development of colorectal cancer, which occurred respectively in 133 and 136 women from the treatment and control groups RR = 0.97 (95% CI, 0.77–1.24) [83].

##### Long-Term Effect of Aspirin

In the case of aspirin, long-term use studies have also been performed. Because of increased risk of vascular events, long-term use (more than 5 years) of COX-2 inhibitors in prevention is rather not feasible, in the contrary of aspirin use [42]. The trials of aspirin in prevention of vascular events were followed up to establish the effect of aspirin on risk of colorectal cancer over 20 years during and after the trails by Rothwell’s team [42]. They found that aspirin might have a greater effect on cancer of the proximal than distal colon or rectum. They also shown the same effect for 75 mg of aspirin daily as for higher doses, in contrary to very low doses (30 mg per day) which occurred to be ineffective. It was also established that long-term use of the drug caused reduction in fatal colorectal cancer (more than reduction in incidence) [42]. Rothwell’s team determined also the effect of allocation to aspirin on risk of cancer death in relation to scheduled duration of trial treatment for gastrointestinal and non-gastrointestinal cancers [84]. They found that the use of aspirin for at least 5 years is required before reductions in risk of cancer are observed. It reduced the 20-year risk of death due to all studied solid cancers and gastrointestinal cancer, but not hematological cancer. The reduction of death due to cancer by the long-term use of the drug was 20% and was limited to certain cancers, most particularly adenocarcinomas [84]. Finally the Rothwell team assessed whether any weight or height dependence was evident for the effect of aspirin on 20-year risk of colorectal cancer [85]. They found that 75–100 mg aspirin once a day was an ineffective in reduction of cardiovascular events, sudden cardiac death and cancer in people weighing 70 kg or more, particularly in those who smoked or were treated with enteric-coated formulations [85].

#### 2.7.2. Sulindac in Chemoprevention of Colorectal Cancer

Although cases of colorectal cancer are mostly sporadic, about 5% of them are caused by hereditary syndromes, such as FAP and Lynch syndrome, also known as HNPCC. FAP is a genetic disease caused by a mutation in the APC gene inherited in an autosomal dominant pattern. This syndrome results in the development of numerous adenomas in the large bowel. FAP is characterized by the early onset of symptoms (polyps appear as early as the teenage years) and almost 100% lifetime risk of developing colorectal cancer. Lynch syndrome, also an autosomal dominant disorder, results from a germline mutation in one of the mismatch repair genes, being DNA repair genes. Lynch syndrome may predispose not only to the development of colorectal cancer but also to other organ cancers, especially of the genitourinary system. The estimated lifetime risk of developing colorectal and endometrial cancer in individuals with HNPCC is 50% and 40%, respectively. Given the high risk of developing colorectal cancer, patients who are at risk of these hereditary syndromes are a valuable group of candidates for clinical trials with chemopreventive agents [29].

Between 1993 and 2001, the study on young patients aged 8–25 years with APC mutation, yet attenuated FAP phenotype, who had no adenomatous polyps identified in the large intestine, was carried out. The study included 41 participants, randomly assigned to a group receiving sulindac (*n* = 21) twice a day or a placebo group (*n* = 20). The sulindac dose was body weight-dependent at the start of the study and was 75 mg for subjects weighing 20–44 kg (*n* =11) and 150 mg for subjects weighing over 44 kg (*n* = 10). The study lasted four years, and at the end of the study, all but three patients were on a dose of 150 mg. The development of adenomatous polyps was evaluated by sigmoidoscopy before the beginning of the study and then every four months throughout the trial. During each endoscopy, the number of polyps on the circumference of the colorectum from 20 cm to the anal verge was counted, and their diameter was measured. Each study was video-recorded. Towards the end of four years, five patients from the treatment group and six from the control group withdrew from the study. The number of patients who developed one or more adenomas did not differ significantly between the groups. In the treatment group, adenomas developed in nine out of 21 patients (43%), while in the control group, in 11 out of 20 patients (55%). Moreover, there were no significant differences between the groups in terms of both quantity and diameter of polyps during the treatment period lasting 40 months or more. On the other hand, the levels of prostaglandins D_2_, E_2_, F_2α_, and TXA_2_ in the colorectal mucosa were significantly lower in patients receiving sulindac, which could be evidence of adherence to treatment recommendations. In summary, the standard sulindac doses assessed in this clinical trial did not prevent adenomas from developing in young patients with FAP who did not have the lesions in the colon at the beginning of the trial [78].

#### 2.7.3. Celecoxib in Chemoprevention of Colorectal Cancer

In the period between 1999 and 2004, a clinical trial called APC was conducted to assess the safety and efficacy of celecoxib (at daily doses of 400 mg and 800 mg) to prevent colorectal adenomas from recurring. The patients recruited for this clinical trial had a history of recurrent adenomas or had an adenoma larger than 5 cm in diameter. Their age was between 31 and 88 years. Individuals with hereditary syndromes (FAP and HNPCC) and inflammatory bowel diseases were excluded from the trial. Before the start of the experiment, the participants underwent a colonoscopy with polypectomy. For the study, a total of 2035 participants were randomly assigned to three groups: celecoxib 2 × 400 mg daily (*n* = 679), celecoxib 2 × 200 mg daily (*n* = 685), and placebo (*n* = 671). The intervention was planned for three years, but it was discontinued in December 2004 before completion due to an increasing number of cardiovascular adverse events. In the first year, colonoscopy was performed in 89.5% of patients while in 75.7% of patients in the third year. The estimated cumulative incidence of one or more adenomas in the third year was 60.7% in the placebo group, 43.2% in the group with celecoxib 400 mg daily, and 37.5% in the group with celecoxib 800 mg per day. The study results confirm the efficacy of celecoxib in reducing the risk of colorectal adenomas. However, it is not an agent for routine use in chemoprevention due to the mentioned risk of adverse events. Nevertheless, it may be a valuable option for patients at risk of colorectal cancer but with a low risk of cardiovascular adverse events [80].

Between 2001 and 2005, a similar clinical trial to the APC study was conducted, called Prevention of Colorectal Sporadic Adenomatous Polyps (PreSAP). This experiment also assessed the efficacy of celecoxib in preventing the recurrence of adenomas in the colon in high-risk individuals with sporadic adenomas (at least one >6 mm or 2–10 of any size) by colonoscopy. The inclusion criteria were the age of over 30 years and no incidence of a hereditary syndrome associated with colorectal cancer (FAP and HNPCC), and no incidence of inflammatory bowel disease. Excluded were also patients with a previous history of invasive cancer in the last five years. All adenomas had been surgically removed before the examination. A total of 1561 individuals participated in the trial. Following randomization, the study group included 933 individuals, while 628 patients were in the control group. In contrast to the APC study, this experiment assessed only the effect of celecoxib at a lower daily dose (2 × 200 mg). The follow-up colonoscopy was performed after one year and three years after that. The primary endpoint was detecting one or more adenomas in patients from the study group compared to the control group. The cumulative incidence of adenomas in the third year was 33.6% in the celecoxib group and 49.3% in the placebo group. Regarding the advanced adenomas detection, the cumulative incidence rate was 5.3% and 10.4%, respectively, in the study and control group. Therefore, celecoxib 400 mg per day is considered to significantly reduce the risk of colorectal adenomas recurrence in post-polypectomy subjects [79].

Also, a multicenter Children’s International Polyposis (CHIP) study was conducted in children (10–17 years) with familial adenomatous polyposis (FAP) to assess the safety and efficacy of celecoxib in preventing the development of colorectal polyps compared to placebo during the five-year treatment. In the years between 2006 and 2013, participants from thirteen countries were recruited. A total of 106 patients were randomized; 55 patients were assigned to the group receiving celecoxib, and 51 to the placebo group. The celecoxib daily dose in the study group was body-weight dependent: 2 × 200 mg (from 25 to 37.5 kg), 2 × 300 mg (from 37.6 to 50 kg) or 2 × 400 mg (>50 kg). To qualify for the study, patients could not have more than 20 polyps >2 mm in size. All polyps >2 mm were removed before the intervention started. During the study, a colonoscopic examination was performed annually, and all polyps >2 mm, if found (if less than 20), were removed. The study assumed a maximum five-year treatment period. However, it was completed in 2013, and the median duration of the intervention was 23 months in the study group and 25.5 months in the control group. The major endpoint was the time to disease progression (TDP), defined as the time from randomization until the occurrence of 20 polyps or more >2 mm in size or colorectal cancer. After the study completion, it was confirmed that none of the participants reached the endpoint of having developed colorectal cancer. The ITT analysis showed 20 patients that reached the endpoint of disease progression (had more than 20 polyps >2 mm), of which seven individuals were from the celecoxib group and thirteen from the placebo group. Thus, progression occurred in 12.7% of participants from the study group and 25.5% of the control group. Moreover, the median time to the progression was 2.1 years in the celecoxib group, while 1.1 years in the group receiving placebo. Therefore, the celecoxib therapy in young FAP patients associates with a decrease in the percentage of individuals who progressed with adenomatous polyposis and a delay in such progression. However, due to the small number of endpoints observed, it is not possible to determine what is the long-term effect of celecoxib therapy on polyp development in children with FAP [81].

#### 2.7.4. Clinical Trials Summary

This paper reviews the clinical trials that examine the chemopreventive properties of selected NSAIDs. The procedure descriptions and the results of nine randomized, placebo-controlled, double-blind clinical trials on the use of selected NSAIDs (i.e., acetylsalicylic acid, sulindac, and celecoxib) in colorectal cancer chemoprevention, are presented.

In recent decades, the most developing area of NSAID chemoprevention is that of colorectal cancer. The planned endpoints for most of the presented clinical trials were identical, so the results are comparable.

The majority of trials investigated the use of acetylsalicylic acid, yet the results are not entirely consistent, and the discrepancies may be interpreted in many different ways. In the CAPS studies, a three-year intervention period with ASA 325 mg was applied and proved positive results, namely a 35% reduction in the risk of adenoma recurrence. Similar favorable results were obtained in the Asian population in the clinical trial with ASA 100 mg/day for two years, which recruited patients with both adenoma and colorectal cancer history. However, the study involved a relatively small number of patients [76]. In contrast, the results of the APACC trial are not as clear. Although a one-year treatment reduced the risk of adenoma recurrence by 27%, the final results after four years did not show any benefits of the aspirin, neither at a dose of 160 mg nor 300 mg. Such inconsistency in the final result might be due to a high percentage of patients (over 20%) who withdrew from the study between the first and fourth year, which weakened the statistical power of the final analysis. Another hypothesis to explain these discrepancies assumes that aspirin may show different efficacy depending on the exposure time and polyp history in the individual patient. This hypothesis assumes that aspirin at a daily dose of 300 mg may significantly inhibit the recurrence of polyps thanks to its antiproliferative effect. However, more prolonged exposure to ASA would be required for the optimum chemopreventive effect (i.e., prevention of new polyp formation) [75]. However, such a conclusion is disputed by results from the WHS study, where aspirin 100 mg was used every other day for about ten years; no correlation between long-term use and the reduced incidence of colorectal cancer was found. Nonetheless, the treatment population in the WHS experiment included healthy women not checked for colon adenoma presence before randomization. Also, the endpoint differed, being the incidence of colorectal cancer versus adenoma detection in the previously described studies. Further, the 100 mg dose administered every other day could be too low to sustain the most favorable chemopreventive effect [83]. In most of the presented studies, the secondary endpoints were a decrease in total colorectal polyp load in patients receiving ASA. Similarly, a more significant risk reduction was usually observed for the development of advanced adenomas as a secondary endpoint than for the development of adenomas in general, being the primary endpoint. When analyzing the figures, it should also be noted that some small adenomas remain undetected during a routine colonoscopic examination. Consequently, the reduction in the risk of adenoma recurrence following intervention could be less pronounced than the actual one. Therefore, the incidence of advanced adenomas seems to be a more applicable parameter reflecting the actual patient status due to a lower probability of error [86]. In contrast, in the CAPP2 study in patients with Lynch syndrome, the so-called hard endpoint was assigned, which was the occurrence of colorectal cancer. Aspirin 600 mg was used for at least two years. Clinical observation of patients lasted ten years on average, though in some patients, the observation period was extended up to twenty years. Differences in the incidence of colorectal cancer between the groups started to show in the fifth year after starting the treatment. The lower incidence of colorectal cancer in the Aspirin group continued from the fifth year of the observation until the study completion, which would suggest the long-lasting chemopreventive effect, yet occurring with a delay [77]. In conclusion, based on the presented results and evidence from the long-term randomized trials of ASA in the prevention of cardiovascular diseases [42], there is a strong link between the long-term use of the drug and a decrease in the risk of colorectal cancer. However, some issues remain unresolved, and further studies are needed that would focus on the optimum dose for the chemopreventive effect and a long-term (at least 10-year) patient follow-up; the data suggest that the chemopreventive effect of ASA may start to show with a delay.

The number of scientific experiments with sulindac was significantly lower. In the Giardiello et al. study, the population was too small to be considered reliable [78]. The evidence from these studies provides too little data to conclude on the efficacy of sulindac in monotherapy. Nonetheless, the results remain promising enough to continue this course of research.

However, in the case of celecoxib, there is strong evidence for its efficacy in preventing colorectal adenoma recurrence in patients with a history of sporadic colorectal adenomas. Two large studies were conducted, the PreSAP and APC study, on a population of over 1500 patients. The results of both trials, the one assessing the effect of celecoxib at a daily dose of 400 mg and the other one assessing daily doses of 400 and 800 mg, are consistent. The efficacy of celecoxib in the prevention of adenoma recurrence increases in a dose-dependent manner: at a dose of 800 mg, the risk of recurrence decreases by almost half. Nevertheless, celecoxib 400 mg reduces the risk by over 30% [79,80]. The results of the CHIP study involving children with FAP also support the efficacy of celecoxib in reducing the risk of adenoma development, yet further studies on a larger population are needed to confirm these findings [81].

#### 2.7.5. NSAIDs in Clinical Trials—Perspectives

In addition to experiments aimed at the continuation of previous studies, entirely new clinical research projects with NSAIDs are currently being developed. The search of the world-wide clinical trials database ClinicalTrials.gov (https://www.clinicaltrials.gov) produced insights into the directions of current and future research. Table 3 presents the data collected.

Multiple clinical trials are currently underway, assessing the use of aspirin not only in the primary but also secondary and tertiary chemoprevention. A large ADD-ASPIRIN trial is being conducted, assessing the effects of long-term aspirin administration on the frequency of recurrence of common types of cancer and disease-free survival. Most of the current research focuses on expanding the knowledge of colorectal cancer. The trial with sulindac and DFMO is a factorial design trial this time that will enable an objective assessment of the effect of each drug in monotherapy. In addition to the data presented in Table 3, many non-randomized open clinical trials with celecoxib have been identified. Efficacy of celecoxib used as adjuvant therapy to anticancer drugs, namely cytostatic agents (e.g., gemcitabine, cisplatin, fluorouracil, cyclophosphamide) or monoclonal antibodies, in various types of tumors has been assessed.

## 3. Conclusions

Chemoprevention is a relatively new research field, although prevention, early diagnosis, surgical and adjuvant treatment should constitute a complete anticancer strategy. Cancer prevention should involve a healthy lifestyle and diet, combined with chemoprevention in higher-risk healthy individuals or patients with precancerous lesions, and those at risk for a second primary cancer. In the future, the complete anticancer strategy will probably evolve into more sophisticated methods of management of higher-risk individuals and a more personalized approach.

The use of NSAIDs as chemopreventive agents in patients with CRC seems to be an attractive option because of their effect not only on tumor molecular biology but also on the systemic and local inflammatory response, combined with the low cost of adjuvant therapy. The clinical trial results confirm the efficacy of NSAIDs in reducing the risk of colorectal adenomas. However, chemoprevention designed as a continuous, long-term treatment may cause adverse clinical effects, therefore less toxic compounds with well-defined safety profiles need to be identified. Future studies should also clarify the exact role of NSAIDs in the management of patients with colorectal cancer. Hopefully, this new approach would also change the perception of colorectal cancer from a terminal disease to the chronic one that can be taken under control.

## Figures and Tables

**Figure 1 cancers-13-00594-f001:**
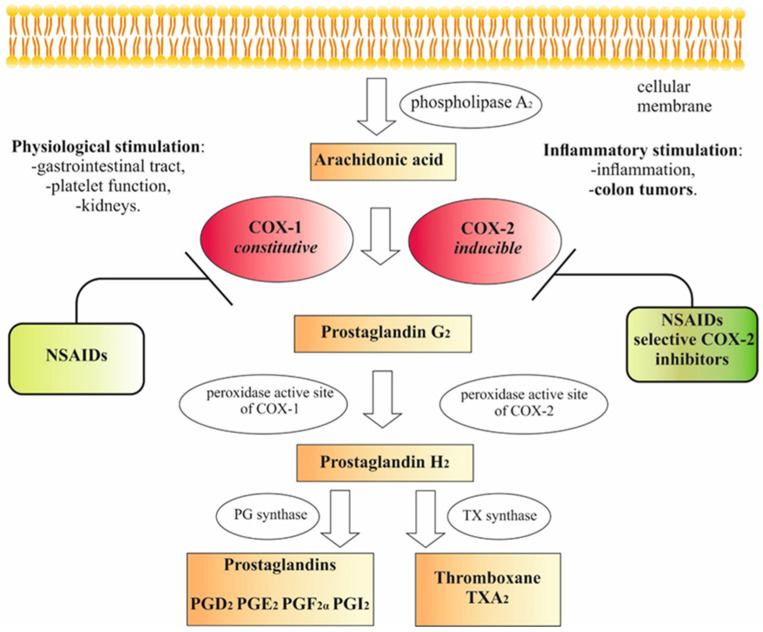
Effect of NSAIDs on eicosanoid synthesis pathways [35,36,37].

**Table 1 cancers-13-00594-t001:** The examples of the NSAIDs drugs

**Aspirin**	**Ibuprofen**
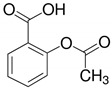	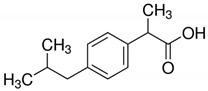
**Meloxicam**	**Naproxen**
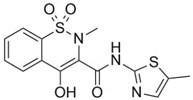	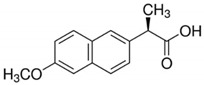
**Sulindac**	**Celecoxib**
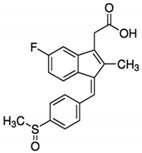	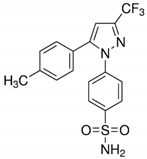

**Table 2 cancers-13-00594-t002:** Summary of the most important data from clinical trials of NSAIDs in colorectal cancer chemoprevention.

Clinical Trial, Source, Year	Study Drug	Number of Patients, *n*	Drug Dose, Treatment Period	Population	Endpoint	Result
CAPS, [74], 2003	ASA	635 (317—ASA; 318—placebo)	ASA 325 mg for 3 years	patients with a history of colorectal cancer without relapse for at least 5 years after tumor resection; age 30–80 years	at least one colon adenoma in the study group vs. control group	RR = 0.65 95% CI (0.46–0.91)
APACC, [75], 2012	Lysine acetylsalicylate (soluble acetylsalicylic salt)	272 (73—lower dose of ASA; 67—higher dose of ASA; 132—placebo)	ASA 160 mg or 300 mg for 4 years	patients with a history of sporadic colon adenomas; age 18–75 years	at least one colon adenoma in the study group vs. control group after one year	RR = 0.73 95% CI (0.52–1.04)
					and after four years	RR = 0.96 95% CI (0.75-1.22)
Ishikawa et al. [76], 2014	ASA	311 (152—ASA; 159—placebo)	ASA 100 mg for 2 years	patients with single or multiple adenomas and/or colorectal adenocarcinomas, originating from Japan; age 40–70 years	frequency of relapse in the study group vs. control group	OR = 0.60 95% CI (0.36–0.98)
CAPP2, [77], 2020	ASA; resistant starch	861 (427—ASA; 434—placebo)	ASA 600 mg; resistant starch 30 g for at least 2 years	patients with Lynch syndrome; min age 45 years	colorectal cancer onset	HR = 0.65 95% CI (0.43–0.97)
Giardiello et al. [78], 2002	Sulindac	41 (21—Sulindac; 20—placebo)	2 × 150 mg or 2 × 75 mg for 4 years	young patients with FAP; age 8–25 years	colon adenomas	no positive results; adenomas developed in 43% of patients on sulindac and 55% of patients on placebo
PreSAP, [79], 2006	Celecoxib	1561 (933—Celecoxib; 628—placebo)	Celecoxib 400 mg daily for 3 years	patients with a history of sporadic colon adenomas; age > 30 years	at least one colon adenoma in the study group vs. control group	RR = 0.64 95% CI (0.56–0.75)
					detection of advanced adenomas	RR = 0.49 95% CI (0.33–0.73)
APC, [80], 2006	Celecoxib	2035 (679—higher dose of Celecoxib; 685—lower dose of Celecoxib; 671—placebo)	2 × 200 mg or 2 × 400 mg of Celecoxib daily for 3 years	patients with a history of sporadic colon adenomas; age 31–88 years	at least one colon adenoma in the study group vs. control group	in a group on a lower dose: RR = 0.67 95% CI (0.59–0.77); in a group on a higher dose: RR = 0.45 95% CI (0.33–0.63)
					advanced adenomas	in a group on a lower dose: RR = 0.43 95% CI (0.31–0.61); in a group on a higher dose: RR = 0.34 95% CI (0.24–0.50)
CHIP, [81], 2017	Celecoxib	106 (55—Celecoxib; 51—placebo)	weight-dependent dose: 2 × 200 mg (25.0–37.5 kg); 2 × 300 mg (37.6–50.0 kg); 2 × 400 mg (>50.0 kg) for max. 5 years	young patients with FAP; age 10–17 years	time to disease progression (TDP), defined as development of 20 polyps or more >2 mm by colonoscopy	median in the study group—2.1 years; median in the control group—1.1 years
					the percentage of patients with progression	12.7% of patients in the study group and 25.5% of patients in the control group

RR: relative risk, OR: odds ratio, HR: hazard ratio, CI: confidence interval.

**Table 3 cancers-13-00594-t003:** Current randomized, controlled, double-blind clinical trials to assess NSAIDs in cancer chemoprevention based on https://www.clinicaltrials.gov.

Clinical Trial Name/ Identification Number	Study Drug (Dose)	Chemoprevention	Population
ADD-ASPIRIN/ NCT02804815	ASA (100/300 mg)	cancer recurrence prevention: colorectal, breast, prostate, stomach, esophageal	patients with past cancers
ASAMET/ NCT03047837	ASA (100 mg) + Metformin (850 mg)	prevention of colorectal cancer progression	patients with colorectal cancer stage I–III
APREMEC/ NCT02607072	ASA (100/200 mg)	prevention of colorectal cancer progression	patients with colorectal cancer
ASAC/ NCT03326791	ASA (160 mg)	prevention of metastasis recurrence	patients with colorectal cancer and metastasis to the liver
AAS-Lynch/ NCT02813824	ASA (100/300mg)	colorectal cancer prevention	patients with Lynch syndrome
NCT02965703	ASA	colorectal cancer prevention	patients with colorectal adenomas
NCT02052908	Naproxen	colorectal cancer prevention	patients with Lynch syndrome
PACES/ NCT01349881	Sulindac (150 mg) and/or DFMO (500 mg)	prevention of colorectal cancer recurrence	patients with past colorectal cancer

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
