# Peer review of "Non-Steroidal Anti-Inflammatory Drugs in Colorectal Cancer Chemoprevention"

_cancers, 2021, doi:10.3390/cancers13040594_

Round 1

Reviewer 1 Report

Comments to the Author

In the review article titled “Non- steroidal anti-inflammatory drugs in colorectal cancer chemoprevension”. The authors described the current state of scientific knowledge on the potential benefit of the use of Non- steroidal anti-inflammatory drugs (NSAIDs (such as Aspirin, Sulindac and Celecoxib) as chemopreventive agents in patients at risk of colorectal cancer. They tried to make a link between the long-term uses of the NSAIDS and decrease in the risk of colorectal cancer through extensive research on available literature.

Overall, the study is interesting and I am sure it will be an important addition in the colorectal cancer management globally. However, the authors need to thoroughly check the article for grammatical and typo errors to make it coherent and comprehensive.

Therefore, for these reasons and other issues listed below, I suggest reconsidering this article for publication in cancers after minor revision.

Issues and Comments:

  1. I think author should remove “simple” from line 11.
  2. Author should replace Rewiev with Review in line 68.
  3. I would appreciate if author could add one or two paragraph related to adverse clinical effects after the long-term use of NSAIDS. I think author can get some insight from the article published by Takahiro etal “Effects of NSAIDS on the risk factors of colorectal cancer: a mini review”.

Author Response

All changes to the manuscript text are marked in yellow colour.

  1. I think author should remove “simple” from line 11.

Response: „Simple Summary” comes from Cancer’s Microsoft Word template. We are not sure if we can remove it.

  1. Author should replace Rewiev with Review in line 68.

Response: Thank you, the spelling mistake has been corrected.

  1. I would appreciate if author could add one or two paragraph related to adverse clinical effects after the long-term use of NSAIDS. I think author can get some insight from the article published by Takahiro etal “Effects of NSAIDS on the risk factors of colorectal cancer: a mini review”.

Response: Thank you very much for this hint. As we are pharmacists, the side effects of NSAIDs are obvious to us. However, it is important to mention the subject for readers who are not specialists in the field of pharmacology or medicinal chemistry. To meet your demand, short paragraph have been added as follows:

As bioactive lipids, prostanoids activate specific cell membrane receptors (e.g. prostaglandins, prostacyclin I2, and thromboxane A2 receptors ) related to specific biological functions, therefore, inhibition of their synthesis is associated with the occurrence of adverse clinical effects after the long-term use of NSAIDs [Patrono 2019]. COX inhibition can increase the risk of gastrointestinal ulcers and bleedings, it also interfere with kidney function [Hamoya 2016]. The selective COX-2 inhibitors (such as Celecoxib and Rofecoxib) were intended to lower the rates of gastrointestinal adverse events but are associated with cardiovascular risk (e.g. stroke or heart failure) [Gislason 2020]. One of the proposed mechanisms for this risk of NSAIDs is the observed shift in the prothrombotic-antitrombotic balance on endothelial surface towards thrombosis. The simplified hypothesis of an association between the degree of COX-2 inhibition and the cardiovascular risk is that the more COX-2 inhibition that an NSAID exerts relative to COX-1 inhibition, the higher the risk of cardiovascular events is [Gislason 2020]. Because of increased risk of vascular events, long-term use (more than 5 years) of COX-2 inhibitors in prevention is rather not feasible (more feasible is Aspirin use) [Rothwell 2010]. Low-dose Aspirin (COX-1 inhibitor) therapy inhibits TXA2 –dependent platelet function, that is why patients are taking it to prevent myocardial infarction, but antiplatelet effect of this drug might also increase the risk of bleeding during a subsequent operation [Spiegel 2020]. NSAIDs may also impair the therapeutic benefits of ACE inhibitors and β-blockers by inhibiting prostaglandins biosynthesis in the kidneys, which results in impaired vasodilatation, decreased renal function, sodium and water retention which may cause oedema [Patrono 2019].

Reviewer 2 Report

The manuscript by Maniewska and Jeżewska is a review paper which analyze the preclinical and clinical significance of various non-steroidal anti-inflammatory drugs in colorectal cancer chemoprevention. In my opinion, this is a well-designed and performed study. The literature review offers a useful overview of current research and policy, and the resulting bibliography provides a very useful resource for current practitioners. The manuscript is written very well and I do not find any significant incorrectness.

My following comments are of minor character:

It is evident that the manuscript is totally unformatted. I don't know the reason, but you should correct it.

Line 20: To avoid repeating “cancer”, you can use “malignancies” (end of first line).

Line 38: ….. being the world’s fourth deadly cancer. Here a reference is needed.

Line 70: “also known” or “commonly known” instead of also commonly known.

Line 95: A space is needed after the point in “effects [34].The most”.

Lines 114-119: These sentences are not clear and hard to follow. Please rephrase/rewrite them.

Line 147: A better table title could be: Examples of NSAID drugs

Line 154: ….. stroke and heart attack effectively. Here a reference is needed.

Line 173: The role of COX-2 in the process of carcinogenesis describes section 2.3 ???? Maybe the authors want to say, “is described” or “has been described”?

Table 2: This is a very valuable table. However, the table is not correctly formatted and, therefore, is complicated to follow the content of the table. Please amend it.

Author Response

All changes to the manuscript text are marked in yellow colour.

  1. Line 20: To avoid repeating “cancer”, you can use “malignancies” (end of first line).

Response: Thank you for this hint. We have followed your suggestion.

  1. Line 38: ….. being the world’s fourth deadly cancer. Here a reference is needed.

Response: Thank you for this suggestion. We have added a bibliographic reference.

  1. Line 70: “also known” or “commonly known” instead of also commonly known.

Response: Thank you for this suggestion. We have followed it and replaced the phrase "also commonly known" with "commonly known".

  1. Line 95: A space is needed after the point in “effects [34].The most”.

Response: Thank you for this editorial remark, we have added a space in the appropriate part of the text.

  1. Lines 114-119: These sentences are not clear and hard to follow. Please rephrase/rewrite them.

Response: Thank you for this hint. To meet your demand, we have rewrote those sentences as follows:

The activated transcription factors are present in the inflammatory microenvironment. They stimulate the expression of various cytokines, chemokines, COX-2, and, consequently, prostaglandins (inflammatory mediators). As a result of these pro-inflammatory substances being released, inflammatory cells accumulate in the surrounding tissue.

  1. Line 147: A better table title could be: Examples of NSAID drugs

Response: Thank you for this hint. We have followed your suggestion and changed the table title.

Line 154: ….. stroke and heart attack effectively. Here a reference is needed.

Response: Thank you for this suggestion. We have added a bibliographic reference.

  1. Line 173: The role of COX-2 in the process of carcinogenesis describes section 2.3 ???? Maybe the authors want to say, “is described” or “has been described”?

Response: Due to the fact that English is not our mother tongue, we did not notice that the sentence written in this way is grammatically incorrect, so we are grateful that you pointed it out. We have rewrote this sentence as follows: „The role of COX-2 in the process of carcinogenesis has been described in section 2.3.”

  1. Table 2: This is a very valuable table. However, the table is not correctly formatted and, therefore, is complicated to follow the content of the table. Please amend it.

Response: Thank you for noticing – something went wrong while uploading the manuscript. Table formatting was lost while uploading to the journal website. The table has been designed to cover the entire page of the journal, we hope that it will be correctly placed this time.